# Cardiorespiratory Fitness and Physical Activity among Children and Adolescents: 3-Year Longitudinal Study in Brazil

**DOI:** 10.3390/ijerph191811431

**Published:** 2022-09-11

**Authors:** Diego Augusto Santos Silva, Eliane Cristina de Andrade Gonçalves, Emerson Filipino Coelho, Matheus Santos Cerqueira, Francisco Zacaron Werneck

**Affiliations:** 1Núcleo de Pesquisa em Cineantropometria e Desempenho Humano, Departamento de Educação Física, Centro de Desportos, Universidade Federal de Santa Catarina, Florianópolis 88040-900, SC, Brazil; 2Facultad de Ciencias de la Salud, Universidad Autónoma de Chile, Providencia 7500912, Chile; 3Departamento de Educação Física, Universidade Estadual de Maringá, Maringá 87020-900, PR, Brazil; 4Laboratório de Estudos e Pesquisas do Exercício e Esporte, Escola de Educação Física, Universidade Federal de Ouro Preto, Ouro Preto 35400-000, MG, Brazil; 5Instituto Federal de Educação, Ciência e Tecnologia do Sudeste de Minas Gerais-Campus Rio Pomba, Rio Pomba 36180-000, MG, Brazil

**Keywords:** adolescent health, exercise, physical fitness

## Abstract

The aim of this study was to investigate the effect of physical activity (PA) levels over 36 months on the cardiorespiratory fitness (CRF) of children and adolescents. This was a longitudinal study carried out from 2016 to 2019 with 127 children and adolescents (82 males and 45 females) aged 13.5 ± 1.2 years at baseline. The 20 m shuttle run test (20 mSRT) was used to predict CRF. The questionnaires PAQ-C and PAQ-A were used to investigate PA. The 2 × 4 repeated measures ANCOVA was used, and the significance level was *p* ≤ 0.05. There was a linear trend in the interaction between level of PA and CRF throughout the 36-month study period in both sexes (males—distance covered: F1.78 = 4.567; *p* = 0.04; VO_2max_: F1.78 = 5.323; *p* = 0.02; females—distance covered: F1.41 = 6.989; *p* = 0.01; VO_2max_: F1.41 = 6.585; *p* = 0.01). Physically active males showed a gradual increase in CRF throughout the analyzed period. For females, CRF remained constant in the first 24 months, showing a slight increase only after 36 months in physically active ones. The findings of this study reinforce the importance of PA throughout childhood and adolescence to improve CRF.

## 1. Introduction

Aerobic fitness reflects, among other things, the general ability of the cardiovascular and pulmonary systems to supply oxygen during exercise [1]. Thus, it is considered a central element of health [2]. In adulthood, low levels of aerobic fitness are related to cardiovascular diseases, early death from all causes, diabetes, and stroke [3]. In childhood and adolescence, adequate levels of aerobic fitness can reduce obesity rates [4], promote healthy bone development [5], better academic performance [6] and mental health [7], decrease the amount of total and central body fat [8,9,10,11,12], and reduce the risk of developing cardiovascular diseases in adulthood [13].

In recent decades (from 1981 to 2014), the aerobic fitness of children and adolescents has decreased, with a reduction of 7.2% around the globe [14]. In Brazil, approximately 30% of children and adolescents have aerobic fitness considered adequate for health parameters [15,16]. Some factors negatively influence levels of aerobic fitness such as biological factors (male gender), socioeconomic factors (economic level) and modifiable lifestyle factors (excess body fat, screen time and physical activity) [12].

Considering that physical activity and aerobic fitness are variables related to physical and mental health indicators (especially modifiable factors), studies have focused on identifying whether there is an association between them [12,17,18]. In addition to the fact that most studies [12] have a cross-sectional design, which makes it impossible to identify cause and effect, findings seem to be inconclusive. While some isolated studies found no association between physical activity and aerobic fitness [17,18], a systematic review presented the synthesis of different studies, demonstrating that there is a positive association between variables [12]. However, a systematic review [12] presented evidence only from studies with a cross-sectional design, which limits the inferences of the results.

The divergent results impose limitations on the planning of health promotion actions, since it is not known with certainty whether physical activity actually interferes with aerobic fitness parameters and how this relationship occurs. Therefore, this fact makes it difficult to design interventions aimed at encouraging the regular practice of physical activity in order to improve aerobic fitness and, consequently, the health of adolescents. Thus, this study periodically evaluated (for 36 months) the practice of physical activity in order to analyze whether the maintenance or increase in this practice over time could increase the levels of aerobic fitness of children and adolescents. With these results, it will be possible to develop specific and effective interventions at a community and school level in order to stimulate the practice of physical activity in the young community as a way to improve the levels of aerobic fitness and consequently the health of this population. Thus, the aim of this study was to analyze the effect of physical activity over 36 months on the levels of aerobic fitness of children and adolescents.

## 2. Materials and Methods

This study is part of the “*Projeto Atletas de Ouro*: Multidimensional and Longitudinal Assessment of the Sports Potential of Young Athletes”, approved by the Research Ethics Committee of the Federal University of Ouro Preto (CAAE: 32959814.4.1001.5150).

### 2.1. Population and Sample

The target population was composed of students enrolled in the ‘Military School of Brazil’ system. For convenience, the study was carried out at the Military School of Juiz de Fora (CMJF), Minas Gerais, Brazil, at which approximately 900 students attend from Basic Education-Elementary School (6th to 9th grade) to High School. Most students are children of military personnel in the Armed Forces, and there are also students who enter through public contest held in the 6th year of elementary school and 1st year of high school. In the cross-sectional stage of this study, all students that enrolled in 2016 at CMJF were invited to participate in the research (412 adolescents (216 males and 196 females) aged 11–16 years). In the longitudinal stage of the study, these students were reassessed in 2017, 2018 and 2019. With sample losses over the four years, the sample of the longitudinal study consisted of 127 students (82 males and 45 females).

### 2.2. Eligibility Criterion

The inclusion criteria were that participants should be enrolled from the 6th to the 9th year of elementary school. Students who did not submit the Free and Informed Consent Form (FICF) signed by their guardians or those who refused to participate in the research were not evaluated. In addition, students who had any physical or clinical condition that interfered with the performance of physical tests, such as certain types of physical or intellectual disability, were not evaluated. The consent of legal guardians and the assent of students were obtained before participating in the study.

### 2.3. Procedures

Tests were applied during the Physical Education class period, lasting approximately 90 min, on three different days. Data were collected from Monday to Friday between 9:00 am and 12:30 pm. The chronological age of students was determined based on the date of data collection. The evaluation was carried out by properly trained professionals, with fixed evaluators being selected for each test. On the first day, a lecture was held at the CMJF auditorium, in which the test protocol, the collection of sociodemographic data and the application of questionnaire on the level of physical activity were explained, under the supervision of Physical Education teachers. On the second day, anthropometric measurements were collected. On the third day, the 20 m shuttle run test was performed to assess cardiorespiratory resistance. In this test, each evaluator was responsible for controlling the distance covered, the physical state and motivation of three students. The reliability of tests was evaluated through test-retest performed on 20 randomly selected schoolchildren. Intra-class correlation coefficient values > 0.85 were observed.

### 2.4. Instruments

#### 2.4.1. Dependent Variable

Aerobic fitness was assessed by the 20 m shuttle run test [19]. The running pace was set by an audible signal, with initial speed of 8.5 km/h, with an increase of 0.5 km/h at each 1-min interval. The relative VO_2max_ (mL.Kg^−1^.min^−1^) was estimated by the equation: 31.025 + 3.238 × V − 3.248 × I + 0.1536 × V × I, where V: speed in km/h of the last stage reached and I: age in years. For the analysis of the present study, the total distance covered during the test (in meters) was estimated. The variable was continuously considered.

#### 2.4.2. Independent Variable

Physical activity was the main focus of this study. The Physical Activity Questionnaire for Older Children-PAQ-C [20] was applied for participants up to 13 years of age, and for those over 13 years of age, the Physical Activity Questionnaire for Adolescents-PAQ-A [21] was applied. Both instruments assessed different aspects of physical activity in the last seven days and showed adequacy in the translation process, cross-cultural adaptation, reproducibility and concurrent validity for the Brazilian population of the same age group [22].

PAQ-C and PAQ-A questionnaires were composed of questions about the practice of sports and games and physical activities at school and during leisure time, including the weekend. Each question was scored from 1 to 5 and the final score was obtained by the average of questions, representing the range from very less active (1) to very active (5). Scores 2, 3 and 4 indicated slightly active, moderately active and active categories, respectively. Thus, based on the score, individuals were classified as less physically active (scores 1 and 2) and physically active (scores 3, 4 and 5). Similar classifications have already been described in the literature [23].

#### 2.4.3. Characterization Variables

Body mass measurements were performed (digital anthropometric scale with accuracy of 0.05 kg, Welmy, Brazil), height (measuring tape, Sany, Brazil, fixed to the wall, with accuracy of 0.20 cm) and three skinfolds (triceps, subscapularis and leg-scientific adipometer (Sany, Brazil), according to procedures of Norton and Olds [24]. Body mass index was calculated using the following equation: body mass (kg)/squared height (m^2^). Body fat percentage was estimated through the equation of Slaughter et al. [25]. During these measurements, students were wearing Physical Education clothing and were barefoot.

Biological maturation was evaluated by the percentage of predicted adult height reached (%EAP), which was estimated using the Khamis and Roche method [26], using chronological age, current height, and body mass, in addition to the height of biological parents. Based on reference data by age group and sex, maturation stage classifications (delayed, normal, or advanced maturation) were obtained. Parental height was obtained by self-report (mother height: ICC = 0.98 (0.98–0.99); father height: ICC = 0.98 (0.98–0.99)).

### 2.5. Statistical Analysis

Data were described as mean and standard deviation (quantitative variables) and percentages (qualitative variables). Student’s t-test for independent samples was used to test for differences between less active and active groups. The Chi-square test was used to test associations between qualitative variables. When comparing means, the effect size was evaluated by Cohen’s d, adopting the classification proposed by Cohen [27]. The reproducibility of measurements was evaluated by the intra-class correlation coefficient (ICC). To investigate the effect of the level of physical activity on the aerobic fitness evolution from 2016 to 2019, 2 × 4 repeated measures ANCOVA was used. Chronological age and aerobic fitness at baseline were used as covariates. The assumptions of data normality and sphericity of the variance–covariance matrix were evaluated using the Kolmogorov–Smirnov test and the Mauchly test, respectively. The assumption of the equality of variances was evaluated by Levene’s test. The analysis of the F statistics was performed using the Pillai Trace. When the sphericity assumption was violated, the Huynh-Feldt Epsilon correction factor was used. The main (group and measure) and interaction (group × measure) effects, when statistically significant, were analyzed through multiple comparisons, adopting Bonferroni correction. Analyses were performed separately by sex. All analyses were performed using the IBM SPSS software version 24.0 (IBM Corp., Armonk, NY, USA), adopting *p* ≤ 0.05 for statistical significance.

## 3. Results

Table 1 shows the sample characteristics at baseline in which physically active male adolescents showed greater distance covered in the aerobic fitness test and higher VO_2max_ values compared to less physically active male adolescents (*p* < 0.05). Female adolescents had higher VO_2max_ values compared to those who were less physically active (*p* = 0.04). For the other characteristics, values were similar between physically active and less physically active adolescents. At baseline, the mean levels of physical activity scores for boys (*n* = 216) and girls (*n* = 196) were 2.56 ± 0.64 and 2.36 ± 0.59, respectively (*p* = 0.001; small effect size: *d* = 0.21).

At baseline, the prevalence of physically active and less physically active male students was 38.0% vs. 62.0%, respectively. For females, the prevalence of physically active and less physically active students was 24.5% vs. 75.5%, respectively. No statistically significant association was observed between students’ physical activity classification and age group, both in males (χ^2^ = 3.156; *p* = 0.68; V = 0.12) and females (χ^2^ = 3.638; *p* = 0.60; V = 0.14). Male students presented the following distribution in terms of maturation stage, 32.3% advanced (*n* = 60), 66.1% normal (*n* = 123) and 1.6% delayed maturation (*n* = 3). Among females, 12.6% had advanced (*n* = 22), 75.3% normal (*n* = 131) and 12.1% delayed maturation (*n* = 21). Students’ physical activity classification was not associated with maturational stage in both males (χ^2^ = 3.689; *p* = 0.16; V = 0.14) and females (χ^2^ = 1.533; *p* = 0.46; V = 0.09) (data not shown in tables/figures).

Considering the entire period analyzed, the mean physical activity score in physically active males was higher than compared to those who were less physically active (3.07 ± 0.33 vs. 2.21 ± 0.32; *p* < 0.001; *d* = 2.65, respectively). A similar result was also observed for females (2.98 ± 0.25 vs. 2.09 ± 0.31; *p* < 0.001; *d* = 3.18, respectively). From the practical point of view, the differences observed between groups were of high magnitude (d > 0.80) (data not shown in tables/figures). In Figure 1, it was observed that males had higher physical activity levels than females in all years of the study (Baseline, Male 2.6 ± 0.6 and Female 2.4 ± 0.6 (*d* = 0.33); 12 months, Male 2.6 ± 0.7 and Female 2.3 ± 0.7 (*d* = 0.42); 24 months, Male 2.3 ± 0.7 and Female 2.1 ± 0.6 (*d* = 0.28); 36 months, Male 2.5 ± 0.7 and Female 2.3 ± 0.7 (*d* = 0.28) (*p* < 0.05)).

A statistically significant interaction effect was observed between physical activity and aerobic fitness, suggesting that the change in aerobic fitness over time varied as a function of physical activity. In males, physically active students had a consistent increase in aerobic fitness over time, while in the less active group, this increase occurred only after 36 months. In females, the aerobic fitness of physically active students remained slightly stable in the first 24 months and showed an increase of 11% in relation to the baseline after 36 months, while less active students showed a consistent reduction in aerobic fitness over time. In both males and females, the physically active group showed greater distance covered and greater VO_2max_ compared to the less active group. The description of these data is demonstrated in Table 2. Figure 2 and Figure 3 represent graphically this information.

The trend analysis revealed a linear behavior of the interaction between physical activity and changes in aerobic fitness over 36 months, both in males (Distance covered: F1.78 = 4.567; *p* = 0.04; VO_2max_: F1.78 = 5.323; *p* = 0.02) and females (Distance covered: F1.41 = 6.989; *p* = 0.01; VO_2max_: F1.41 = 6.585; *p* = 0.01). In males, the analysis of stability or tracking of the distance covered in the test suggests that the maintenance of the relative position of each subject within the group over time is lower in the physically active group when compared to the less active group (0.47 (95% CI = 0.29–0.64) vs. 0.75 (95% CI = 0.65–0.84), respectively). In females, stability of aerobic performance was similar between active and less active groups (0.66 (95% CI = 0.40–0.87) vs. 0.64 (95% CI = 0.48–0.78), respectively) (data not shown in tables/figures).

## 4. Discussion

This study presented the effects of physical activity on the change in aerobic fitness in adolescents over 36 months, revealing a linear behavior trend of the interaction between variables during the study period, in both sexes. The findings also showed that physically active boys had a gradual increase in aerobic fitness over the analyzed period, while in the less active group, this increase only occurred after 36 months. Unlike active girls, in which aerobic fitness remained constant in the first 24 months, it showed a slight increase only after 36 months and gradual decrease in less active girls.

Improvements were found in the aerobic fitness of children and adolescents in this study who remained physically active during the 36 months analyzed. Studies that have analyzed the effect of physical activity on maximal oxygen uptake in children and adolescents [28,29,30,31,32] found similar results. A 15-year longitudinal study found that a 30% increase in physical activity scores resulted in a 2% to 5% increase in aerobic fitness levels in adolescents. In addition, studies found that previously untrained children and adolescents who exercised at the appropriate intensity and controlled frequency and duration showed improvements in maximal oxygen uptake (8–9%, 10.3%, 12.2% and 18.9%) [28,29,30,31]. This is because, as the body exercises, especially in endurance training, musculoskeletal adaptations such as improvement in oxidative capacity occur. This adaptation results from the increase in the number of mitochondria, enzymes related to the oxidation of energy substrates, and the rate of fatty acid utilization, which consequently increases maximal oxygen uptake [33]. An interesting fact is that this gradual increase in the percentage of maximum oxygen uptake in the aforementioned studies was associated with the duration of the training period, demonstrating that the longer the training period (12, 15, 28 weeks and 18 months), the greater the increases observed in maximal oxygen uptake [28], which could explain the increase in cardiorespiratory fitness for those who remained active over the 36-month study period.

Biological maturation is a factor that influences the aerobic fitness level of adolescents. This is because VO_2max_ increases in males throughout the period of puberty to adulthood whereas in girls, VO_2max_ increases only at the beginning of puberty, remaining unchanged from the end of puberty to adulthood [8,9,10,11,12]. Thus, it is important to emphasize that biological maturation was tested as a covariate in this study. However, there was no difference between the active and inactive groups in relation to biological maturation and this variable was not correlated with the changes observed over time in aerobic fitness.

The gradual increase in aerobic fitness during the period evaluated in this study occurred in the group of physically active boys, but not in the group of physically active girls. For physically active girls, aerobic fitness remained constant in the first 24 months, showing a slight increase only after 36 months. In general, boys practice higher-intensity physical activities when compared to girls [34] and, for this reason, they may have had significant increases in cardiorespiratory fitness throughout the follow-up. On the other hand, girls tend to practice low-intensity physical activities, which are not sufficient to reach the minimum threshold necessary for significant cardiorespiratory adaptations to occur [34], which may explain the slight stability of aerobic fitness for the group of active girls in the first 24 months of this investigation.

This study provides valuable information on the effect of different levels of physical activity on the aerobic fitness of children and adolescents. Thus, regarding schools, which are important spaces for encouraging the practice of physical activity, these findings suggest a need for more Physical Education teachers or sports club managers to include exercises in Physical Education classes that are designed to ensure physical activities whose duration and intensity are conducive to increase the levels of aerobic fitness. In addition, increasing the weekly frequency of Physical Education classes can promote gains in several health indicators and, consequently, increase the levels of aerobic fitness of children and adolescents. Furthermore, knowing that aerobic fitness provides additional information about the health of children and adolescents [35], which has strong negative associations with cardiometabolic profiles [34], being a factor of protection against cardiovascular risks in adult life [13], it is essential to develop actions that emphasize the practice of routine physical activity also outside the school context to improve the health of all populations, since aerobic fitness is a mediating variable between physical activity and health.

This study has some limitations that should be highlighted. The fact that the sample was collected for convenience may limit the generalization of results. In addition, subjective measures used to analyze physical activity, as in the case of questionnaires, are generally designed to record physical activity at defined periods, as in adults [36]. However, most activities performed by children are sporadic [37], that is, they do not take place in a structured or planned way as in the case of adults, which also prevents the analysis and control of the activity intensity and duration. Therefore, this fact may have influenced the findings on the level of physical activity of children and adolescents in this study.

## 5. Conclusions

The findings of this study showed that the change in aerobic fitness over the 36-month study period was influenced by the students’ level of physical activity. Males who were physically active during the period showed gradual increase in aerobic fitness. For females who were active during the 36 months of analysis, aerobic fitness remained constant in the first 24 months and showed a slight increase at the end of the entire period. For less active girls, the reduction in aerobic fitness was gradual. Future longitudinal studies should be carried out in order to control the intensity of physical activity to identify the minimum intensity of physical activity necessary to have beneficial effects on the aerobic fitness levels of children and adolescents.

## Figures and Tables

**Figure 1 ijerph-19-11431-f001:**
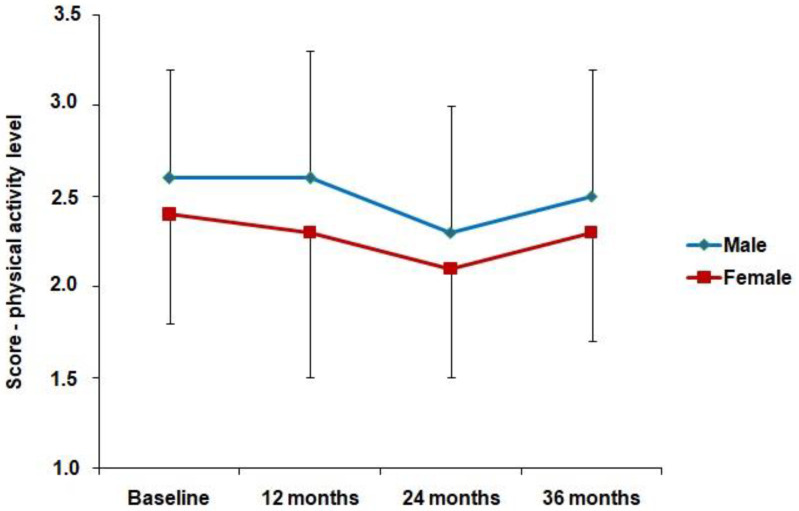
Mean and standard deviation of the physical activity score in males and females over the years of study.

**Figure 2 ijerph-19-11431-f002:**
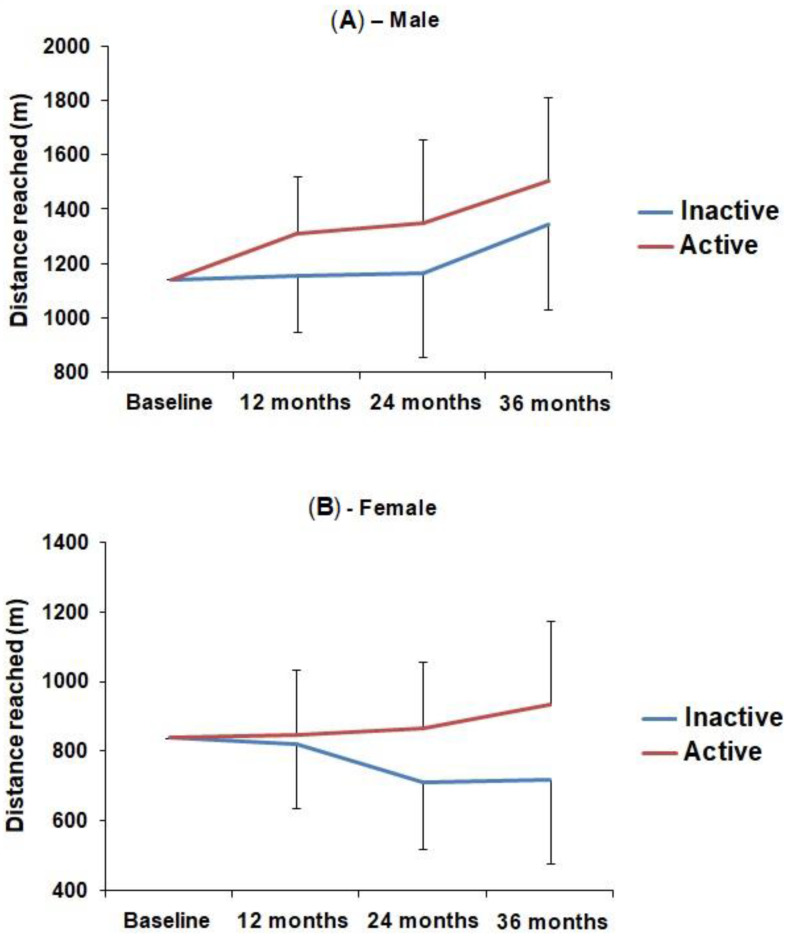
Mean and standard deviation of distance covered in the aerobic fitness test by males (**A**) and females (**B**) according to the level of physical activity over the 36-month study period.

**Figure 3 ijerph-19-11431-f003:**
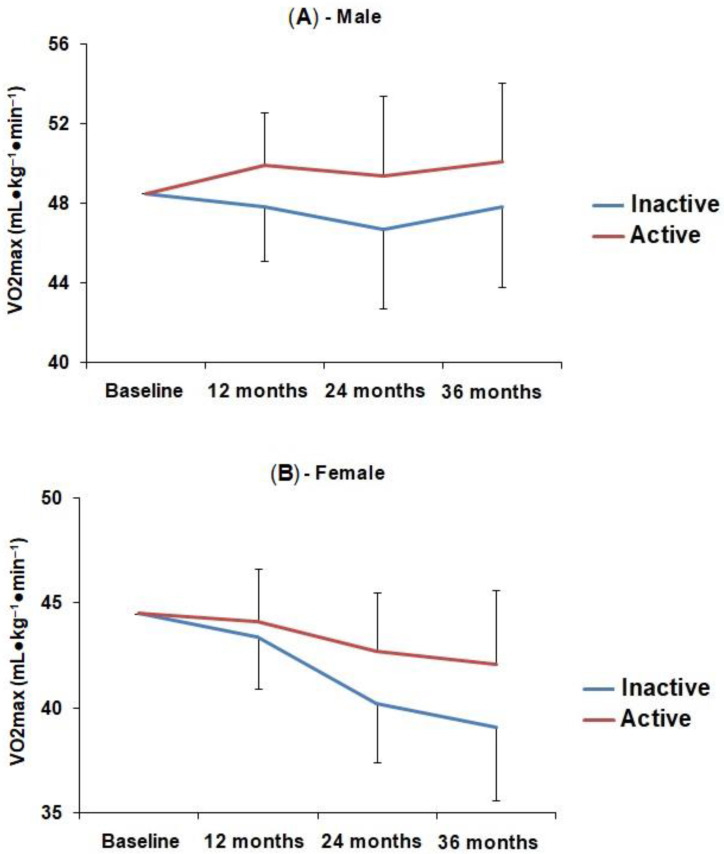
Mean and standard deviation of maximum oxygen uptake in males (**A**) and females (**B**) according to the level of physical activity over the 36-month study period.

**Table 1 ijerph-19-11431-t001:** Descriptive characteristics of the sample of students in relation to the study variables.

	**Less Active** **(*n* = 134)**	**Active** **(*n* = 82)**	***p*-Value**	** *d* **
	**Mean ± SD**	**Mean ± SD**		
**Male**				
Age (years)	13.5 ± 1.3	13.6 ± 1.2	0.58	0.08
Body Mass (kg)	54.8 ± 15.0	55.4 ± 11.5	0.77	0.04
Height (m)	1.63 ± 0.02	1.64 ± 0.01	0.45	0.10
BMI (kg/m^2^)	20.5 ± 4.0	20.6 ± 3.4	0.89	0.02
Fat Percentage (%)	18.2 ± 7.7	16.7 ± 6.9	0.15	0.20
Predicted Adult Height (m)	179.4 ± 5.5	178.9 ± 6.4	0.63	0.08
% EAP (%)	90.9 ± 5.3	91.3 ± 4.6	0.63	0.07
Z score_EAP	0.6 ± 0.7	0.8 ± 0.6	0.19	0.18
Distance covered (m)	1022.8 ± 388.2	1247.8 ± 405.3	<0.001 *	0.58
VO_2max_ (mL.Kg^−1^.min^−1^)	46.7 ± 5.0	49.5 ± 4.8	<0.001 *	0.56
	**Less Active** **(*n* = 148)**	**Active** **(*n* = 48)**	***p*-Value**	** *d* **
	**Mean ± SD**	**Mean ± SD**		
**Female**				
Age (years)	13.5 ± 1.2	13.2 ± 1.3	0.14	0.25
Body Mass (kg)	52.2 ± 11.7	52.3 ± 11.3	0.97	0.01
Height (m)	1.58 ± 0.01	1.57 ± 0.01	0.25	0.21
BMI (kg/m^2^)	20.6 ± 3.6	21.1 ± 3.8	0.48	0.12
Fat Percentage (%)	22.9 ± 6.1	23.0 ± 6.0	0.92	0.02
Predicted Adult Height (m)	164.3 ± 4.0	163.7 ± 5.3	0.46	0.16
% EAP (%)	96.5 ± 3.0	95.7 ± 3.7	0.20	0.24
Z score_EAP	−0.1 ± 1.0	0.1 ± 0.9	0.32	0.17
Distance covered (m)	720.4 ± 266.8	794.1 ± 258.8	0.11	0.28
VO_2max_ (mL.Kg^−1^.min^−1^)	42.4 ± 4.4	44.0 ± 4.2	0.04 *	0.36

SD: standard deviation; * significant difference, *p* < 0.05; BMI: Body Mass Index; % EAP: Percentage of predicted adult height reached; *d*: effect size.

**Table 2 ijerph-19-11431-t002:** Association between aerobic fitness and level of physical activity according to sex over the 36-month study period.

Variable	Group	Mean				Group Effect	Measure Effect	Group * Measure Interaction
		Baseline	12 Months	24 Months	36 Months			
		Mean ± SD	Mean ± SD	Mean ± SD	Mean ± SD			
**Male**						F_1.78_ = 11.383 *p* = 0.001 *	F_3.234_ = 3.878 *p* = 0.01 *	F_3.234_ = 21.250 *p* < 0.001 *
Distance (m)	Active (*n* = 34)	1143.2 ± 0.0	1311.4 ± 211.4	1351.8 ± 306.1	1503.3 ± 312.1			
	Less active (*n* = 48)	1143.2 ± 0.0	1156.5 ± 210.6	1163.7 ± 305.1	1344.7 ± 311.0			
VO_2max_ (mL.Kg^−1^.min^−1^)	Active (*n* = 34)	48.6 ± 0.0	49.9 ± 2.7	49.4 ± 4.1	50.0 ± 4.1	F_1.78_ = 12.811 *p* = 0.001 *	F_3.234_ = 10.049 *p* < 0.001 *	F_3.234_ = 3.863 *p* = 0.01 *
	Less active (*n* = 48)	48.6 ± 0.0	47.8 ± 2.7	46.7 ± 4.0	47.8 ± 4.0			
**Female**						F_1.41_ = 4.546 *p* = 0.04 *	F_3.234_ = 2.795 *p* = 0.04 *	F_3.234_ = 21.250 *p* < 0.001 *
Distance (m)	Active (*n* = 12)	839.5 ± 0.0	849.1 ± 184.6	866.8 ± 200.7	933.4 ± 253.7			
	Less active (*n* = 33)	839.5 ± 0.0	819.7 ± 175.1	711.8 ± 190.4	720.6 ± 240.7			
VO_2max_ (mL.Kg^−1^.min^−1^)	Active (*n* = 12)	44.5 ± 0.0	44.0 ± 2.6	42.7 ± 3.0	42.1 ± 3.7	F_1.41_ = 5.690 *p* = 0.0 *	F_3.234_ = 4.400 *p* = 0.006 *	F_3.234_ = 3.549 *p* = 0.02 *
	Less active (*n* = 33)	44.5 ± 0.0	43.4 ± 2.4	40.1 ± 2.8	39.0 ± 3.5			

SD: standard deviation; * statistically significant difference, *p* < 0.05; Covariates–Males: age = 13.3 years; Distance covered = 1143 m; baseline VO_2max_ = 48.6 mL.Kg^−1^.min^−1^; Females: age = 13.2 years; Distance covered = 839 m; baseline VO_2max_ = 44.5 mL.Kg^−1^.min^−1^.

## Data Availability

The data used to support the findings of this study are available from the corresponding author upon request.

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
