# Peer review of "Cardiorespiratory Fitness and Physical Activity among Children and Adolescents: 3-Year Longitudinal Study in Brazil"

_ijerph, 2022, doi:10.3390/ijerph191811431_

Round 1

Reviewer 1 Report

The research topic is judged to be creative. It is judged that the data collected over the four years was well utilized. However, it is unfortunate that the description of the aerobic program is omitted. Please add a description of the aerobic program. Exercise improves cardiorespiratory endurance for everyone. Although we are talking about aerobic exercise in this study, there is no explanation for aerobic exercise. 

It is judged that appropriate research results were presented by applying various statistical techniques in the research method. Cohen's d value needs an explanation of how it was calculated. The content of the discussion is judged to be general. Differences between this study and previous studies or a creative researcher's argument are required. In conclusion, I hope to include suggestions for follow-up studies.

Author Response

REVIEWER 1

  1. The research topic is judged to be creative. It is judged that the data collected over the four years was well utilized. However, it is unfortunate that the description of the aerobic program is omitted. Please add a description of the aerobic program. Exercise improves cardiorespiratory endurance for everyone. Although we are talking about aerobic exercise in this study, there is no explanation for aerobic exercise. 

Authors: Thank you for the comments. The authors informed that no aerobic exercise interventions were performed in this study. We applied isolated tests as aerobic fitness test and a self-administered questionnaire about the physical activity over 36 months. The description of the tests starts in the “Instruments” part of the text.

Instruments

Dependent variable

            Aerobic fitness was assessed by the 20-m shuttle run test [19]. The running pace is set by an audible signal, with initial speed of 8.5 km/h, with increase of 0.5 km/h at each 1-minute interval. The relative VO2max (ml/kg/min) was estimated by the equation: 31.025 + 3.238*V – 3.248*I + 0.1536*V*I, where V: speed in km/h of the last stage reached and I: age in years. For the analysis of the present study, the total distance covered during the test (in meters) was estimated. The variable was continuously considered.

Independent variable

            Physical activity was the main exposure of this study. The Physical Activity Questionnaire for Older Children - PAQ-C [20] was applied for participants up to 13 years of age, and for those over 13 years of age, the Physical Activity Questionnaire for Adolescents - PAQ-A [21] was applied. Both instruments assess different aspects of physical activity in the last seven days and showed adequacy in the translation process, cross-cultural adaptation, reproducibility and concurrent validity for the Brazilian population of the same age group [22].

            PAQ-C and PAQ-A questionnaires are composed of questions about the practice of sports and games; physical activities at school and during leisure time, including the weekend. Each question has score from 1 to 5 and the final score is obtained by the average of questions, representing the range from very little active (1) to very active (5). Scores 2, 3 and 4 indicate slightly active, moderately active and active categories, respectively. Thus, based on the score, individuals were classified as little physically active (scores 1 and 2) and physically active (scores 3, 4 and 5). Similar classification has already been described in literature [23].”

 REVIEWER 1

  1. It is judged that appropriate research results were presented by applying various statistical techniques in the research method. Cohen's d value needs an explanation of how it was calculated. The content of the discussion is judged to be general. Differences between this study and previous studies or a creative researcher's argument are required. In conclusion, I hope to include suggestions for follow-up studies.

Authors: We thank you for you suggests. First, Cohen's d value was calculated as the difference between two means divided by the mean of the group’s standard deviations. Second, we informed that the information were included in the discussion and conclusion section and are highlighted in yellow.

REFERÊNCIA

Cohen, J. 1988. Statistical Power Analysis for the Behavioral Sciences, 2nd Edition. Routledge.

DISCUSSION

“Improvements were found in the aerobic fitness of children and adolescents in this study who remained physically active during the 36 months analyzed. Studies that have analyzed the effect of physical activity on maximal oxygen uptake in children and adolescents [28-32] found similar results. A 15-year longitudinal study found that a 30% increase in physical activity scores resulted 2% to 5% increase aerobic fitness levels in adolescents. In addition, studies found that previously untrained children and adolescents who exercised at appropriate intensity and controlled frequency and duration showed improvements in maximal oxygen uptake (8-9%, 10.3%, 12.2% and 18.9%) [28-31].”

CONCLUSION

The findings of this study showed that the change in aerobic fitness over the 36-month study period was influenced by the students' level of physical activity. Males who were physically active during the period showed gradual increase in aerobic fitness. For females, girls who were active during the 36 months of analysis, aerobic fitness remained constant in the first 24 months and showed slight increase at the end of the entire period. For little active girls, the drop in aerobic fitness was gradual. So, future longitudinal studies should be carried out in order to control the intensity of physical activity to identify what the minimum intensity of physical activity is necessary to have beneficial effects on the aerobic fitness levels of children and adolescents.

Reviewer 2 Report

Review -Cardiorespiratory fitness and physical activity among children  and adolescents: 4-year longitudinal study in Brazil

The idea of the work is interesting. In my opinion, the work has a lot of understatement. The conducted research is also not original. It is known that the higher the activity level, the better the performance. When analyzing the problem in children, it is important to pay attention to the period of puberty, which changes the values of the VO2max index. The authors didn’t refer to this phenomenon in detail in their work.

Detailed comments:

1.       In the manuscript the authors write: the influence of physical activity. ..lack of information on the form, duration and intensity level of this activity. The question is, has she been monitored in any way?

2.       The author's correspondence address should be related to the institution in which he works

3.       In part of the introduction, the author writes that the decrease in the VO2 max in the elderly depends on the circulatory and respiratory efficiency. There are many other factors that can reduce your physical performance level.

4.       The years 1981-2014 are not five decades

5.       The research problem isn’t new, it is obvious. According to the reviewer, this is only the observation of changes in the examined indicators, in which the results could be predicted. Whoever exercises has a better level of performance even in relation to gender.

6.       Research related to the way of increasing activity or proposing new solutions, encouraging young people to spend time actively is better importance in the development of science.

7.       Why did the authors choose the endpoint of 36 months and what does it mean?

8.       The inclusion criterion provides informations on the analysis of the level of physical fitness of the respondents. How the obtained test results behave in relation to people with different levels of physical fitness.

9.        Statistics and research results are well described. My comments concern the calculated ES, we give it only for changes of p <0.05. Significance levels of changes are missing in the figures.

10.   There is a different notation of the units of measurement of the examined variables in the figures and in the tables

11.    Errors in writing p=0.05 should be p = 0.05, VO2max should be  VO2max

12.    Author Contributions. unbelievable that everyone did everything. They were all study leaders? -needs improvement

13.   No limiting factors in the manuscript

14.    Literature should be corrected according to the journal's guidelines.

Good luck,

Reviewer

Author Response

REVIEWER 2

  1. The idea of the work is interesting. In my opinion, the work has a lot of understatement. The conducted research is also not original. It is known that the higher the activity level, the better the performance. When analyzing the problem in children, it is important to pay attention to the period of puberty, which changes the values of the VO2max index. The authors didn’t refer to this phenomenon in detail in their work.

Authors: Thank you for your comments. The information was included in the text.

“The maturation biological is a factor that influences aerobic fitness level of adolescents. This is because, since VO2max in males increases throughout the period of puberty to adulthood. In girls, VO2max increased only at the beginning of puberty until the end of puberty, remaining unchanged from the end of puberty to adulthood [8-12]. Thus, it is importante to emphasize that biological maturation was tested as a covariate in this study. Although, there was no difference between the active and inactive groups in relation to biological maturation and this variable was not correlated with the changes observed over time in aerobic fitness.” 

REVIEWER 2 

  1. In the manuscript the authors write: the influence of physical activity. ..lack of information on the form, duration and intensity level of this activity. The question is, has she been monitored in any way?

Authors: We thank you for your question. To obtain information about the physical activity we used the PAQ-C and PAQ-A questionnaires. Both instruments assess different aspects of physical activity in the last seven days. So, the physical activity was not systematic monitored. We added in the text: “…PAQ-C and PAQ-A questionnaires are composed of questions about the practice of sports and games; physical activities at school and during leisure time, including the weekend. Each question has score from 1 to 5 and the final score is obtained by the average of questions, representing the range from very little active (1) to very active (5). Scores 2, 3 and 4 indicate slightly active, moderately active and active categories, respectively. Thus, based on the score, individuals were classified as little physically active (scores 1 and 2) and physically active (scores 3, 4 and 5). Similar classification has already been described in literature [23]”.

REVIEWER 2

  1. The author's correspondence address should be related to the institution in which he works

Authors: We thank you the comments and informated that the institution and addressed was included in the text. 

Diego Augusto Santos Silva

Universidade Federal de Santa Catarina – Centro de Desportos, Departamento de Educação Física, Campus Universitário – Trindade – Caixa Postal 476, CEP 88040-900 – Florianópolis, Santa Catarina, Brasil.

Telefone (Fax): +55 48 37218562

REVIEWER 2

  1. In part of the introduction, the author writes that the decrease in the VO2 max in the elderly depends on the circulatory and respiratory efficiency. There are many other factors that can reduce your physical performance level.

Authors: We appreciate the comments. In the new version of the article we added new description.

“Aerobic fitness reflects, among others, the general ability of the cardiovascular and pulmonary systems to supply oxygen during exercise [1]”.

REVIEWER 2

  1. The years 1981-2014 are not five decades.

Authors: We thank for your comments. In the new version this mistake was corrected.

REVIEWER 2

  1. Research related to the way of increasing activity or proposing new solutions, encouraging young people to spend time actively is better importance in the development of science.

Authors: All authors agree with the comment and we inform you that we have included this as a suggestion for further studies. 

“The findings of this study showed that the change in aerobic fitness over the 36-month study period was influenced by the students' level of physical activity. Males who were physically active during the period showed gradual increase in aerobic fitness. For females, girls who were active during the 36 months of analysis, aerobic fitness remained constant in the first 24 months and showed slight increase at the end of the entire period. For little active girls, the drop in aerobic fitness was gradual. So, future longitudinal studies should be carried out in order to control the intensity of physical activity to identify what the minimum intensity of physical activity is necessary to have beneficial effects on the aerobic fitness levels of children and adolescents.”

REVIEWER 2

  1. Why did the authors choose the endpoint of 36 months and what does it mean?

Authors: The study carried out is an integral part of a longitudinal study of annual assessment of health and motor performance indicators in schoolchildren. At the time of conducting the study, we had 4 assessment cohorts that are equivalent to a period of time of 36 months, and it was possible to observe the tracking of the indicators. 

REVIEWER 2

  1. The inclusion criterion provides information on the analysis of the level of physical fitness of the respondents. How the obtained test results behave in relation to people with different levels of physical fitness.

Authors: Thank you for the comments. Statistical analyzes showed measure and time control. The measure control concerns the fact that the heterogeneous sample for physical fitness tests showed this adjustment in the data analysis. That way, if there was any interference in this regard, the statistical data would show. In addition, children and adolescents with physical and intellectual disabilities did not participate in the study. In the new version the information has been changed to be clear.

“The inclusion criteria were that participants should be enrolled from the 6th to the 9th year of elementary school. Students who did not submit the Free and Informed Consent Form (FICF) signed by their guardians or who refused to participate in the research were not evaluated. In addition, students who had any physical or clinical condition that interfered with the performance of physical tests, such as some type of physical or intellectual disability, were not evaluated. The consent of legal guardians and the assent of students were obtained before participating in the study.”

REVIEWER 2

  1. Statistics and research results are well described. My comments concern the calculated ES, we give it only for changes of p <0.05. Significance levels of changes are missing in the figures.

Authors: We thank you for the comments. In the new version of the article the information has been included.

“In Figure 1, it was observed that males had higher physical activity scores than females in all years of study [Baseline, Male 2.6±0.6 and Female 2.4±0.6 (d = 0.33); 12 months, Male 2.6±0.7 and Female 2.3±0.7 (d = 0.42); 24 months, Male 2.3±0.7 and Female 2.1±0.6 (d = 0.28); 36 months, Male 2.5±0.7 and Female 2.3±0.7 (d = 0.28) ((p < 0.05))].”

REVIEWER 2

  1. There is a different notation of the units of measurement of the examined variables in the figures and in the tables.

Authors: We appreciate the reviewer’s comment and attention and inform you that the VO2max unit of measurement has been corrected throughout the text.

REVIEWER 2

  1. Errors in writing p=0.05 should be p = 0.05, VO2max should be VO2max.

Authors: The change was made.  

REVIEWER 2

  1. Author Contributions. unbelievable that everyone did everything. They were all study leaders? -needs improvement.

Authors: The author’s contributions have been redone.

REVIEWER 2

  1. No limiting factors in the manuscript.

Authors: Dear reviewer, the limiting factors found in the last paragraph of the discussion.

“This study has some limitations that should be highlighted. The fact that the sample is for convenience may limit the generalization of results. In addition, subjective measures to analyze physical activity, as in the case of questionnaires, are generally designed to record physical activity at defined periods, as in adults [36]. However, most activities performed by children are sporadic [37], that is, they do not take place in a structured or planned way as in the case of adults, which also prevents the analysis and control of the activity intensity and duration. Therefore, this fact may have influenced the findings on the level of physical activity of children and adolescents in this study.”

REVIEWER 2

  1. Literature should be corrected according to the journal's guidelines.

Authors: The change was made.  

Reviewer 3 Report

The article as a whole is interesting, but there are a number of comments:

1)      Clarify the format of the data representation in Figures 1, 2 and 3.

2)      Data for last paragraph of Results must be added as Supplement file.

3)      Has the correction for multiple comparisons been used? Why didn't the authors use multivariate analysis options?

Author Response

REVIEWER 3

  1. Clarify the format of the data representation in Figures 1, 2 and 3.

Authors: We informed that the figures 1, 2 and 3 represented the mean and standard deviation of data included in table 2. Although, these information has been included in the text for better understand.

“Statistically significant interaction effect was observed between NAF and aerobic fitness, suggesting that the change in aerobic fitness over time varied as a function of NAF. In males, physically active students had consistent increase in aerobic fitness over time, while in the little active group, this increase occurred only after 36 months. In females, the aerobic fitness of physically active students remained slightly stable in the first 24 months and showed increase of 11% in relation to baseline after 36 months, while little active students showed consistent drop in aerobic fitness over time. In both males and females, the physically active group showed greater distance covered and greater VO2max compared to the little active group. The descriptions of these data are demonstrated in table 2. The figure 2 and figure 3 represent graphically this information.” 

REVIEWER 3

  1. Data for last paragraph of Results must be added as Supplement file.

Authors: We appreciate the comment. We understand that this information should be described in the text of the article for better understanding by the reader.

REVIEWER 3

  1. Has the correction for multiple comparisons been used? Why didn't the authors use multivariate analysis options?

Authors: In the longitudinal study, we chose to use ANOVA due to the smaller sample size and the small number of dependent variables (only two). In addition, we included the following information in the statistical analysis:

“The main (group and measure) and interaction (group*measure) effects, when statistically significant, were analyzed through multiple comparisons, adopting Bonferroni correction.”

Round 2

Reviewer 2 Report

Thank you for answers.

After analyzing additional information, I believe that the work does not bring new information to the development of science. Active children will have a higher body capacity at each observation. The authors did not correct the units of measurement according to the SI system. References, information about the corresponding author, Author Contributions do not comply with the guidelines of the journal.

Due to the low scientific level of the manuscript

I propose to reject the manuscript

Author Response

Title: Cardiorespiratory fitness and physical activity among children and adolescents: 3-year longitudinal study in Brazil

Manuscript ID: ijerph- 1852693

Reviewers' Comments

 Reviewer: 2

After analyzing additional information, I believe that the work does not bring new information to the development of science. Active children will have a higher body capacity at each observation. The authors did not correct the units of measurement according to the SI system. References, information about the corresponding author, Author Contributions do not comply with the guidelines of the journal.

Due to the low scientific level of the manuscript

I propose to reject the manuscript.    

Authors: Thank you for your comment. The authors of the article disagree with the reviewer when he claims that the article does not bring any advance to science. In the new version of the article, in the introduction section, we have written that the evidence from a systematic review on the topic brought only studies with a cross-sectional design. As the present study has a longitudinal design, we believe that the design is more appropriate than the design presented in the systematic review studies on the subject.

Regarding the SI system, in the new version of the article we changed the unit of measurement for height - we changed it to meters instead of centimeters. For the distance achieved in the test, we did not change the measurement unit from meters to kilometers because the literature on aerobic fitness field tests adopts the unit of meters in field tests - please check the following articles: Stevinson et al. (2022) and Cordingley et al. (2019). In addition, females reached distances of less than 1 kilometer and the measurement in meters facilitates the reader's interpretation.

In the new version of the article we have adjusted the corresponding author information and the Author Contributions.

References:

- Stevinson C, Plateau CR, Plunkett S, Fitzpatrick EJ, Ojo M, Moran M, Clemes SA. Adherence and Health-Related Outcomes of Beginner Running Programs: A 10-Week Observational Study. Res Q Exerc Sport. 2022 Mar;93(1):87-95. doi: 10.1080/02701367.2020.1799916.

- Cordingley DM, Sirant L, MacDonald PB, Leiter JR. Three-Year Longitudinal Fitness Tracking in Top-Level Competitive Youth Ice Hockey Players. J Strength Cond Res. 2019 Nov;33(11):2909-2912. doi: 10.1519/JSC.0000000000003379.